# JT Gravity on a Finite Lorentzian Strip: Time dependent Quantum Gravity Amplitudes

J. A. ROSABAL [1]

*Departamento de Electromagnetismo y Electrónica, Universidad de Murcia,*
*Campus de Espinardo, 30100 Murcia, Spain.*

We formulate JT quantum gravity on a finite Lorentzian strip. Due to the spatial boundaries of the strip, it is possible to define left and right proper times. With respect to these times we compute non-perturbatively the quantum gravity (QG) time dependent transition amplitude. Lagrangian and Hamiltonian formulations are presented. Special attention is paid to the four corner terms (Hayward terms) in the action that are needed in order to have a well defined variational problem. From a detailed analysis of the gravity boundary condition on the spatial boundary, we find that while the lapse and the shift functions are independent Lagrange multipliers on the bulk, on the spatial boundary, these two are related. This fact leads to an algebraic equation of motion for a particular degree of freedom that is conveniently introduced on the spatial boundaries whose solution can be plugged back into the action allowing to fully determine the time dependent transition amplitude. The final result suggests that time evolution is non-unitary for most of the boundary conditions. Interestingly enough, unitary could be recovered when spatial $AdS_2$ boundary conditions are imposed. Other wave functions for other topologies obtained from the strip by gluing its spatial boundaries are also presented. Remarkably these do not exhibit any non-unitary evolution behavior.

---

[1]jarosabal80@gmail.com

# 1  Introduction

Studying time evolution in QG is particularly challenging. The main reason is due to the lack of time evolution itself. A possible resolution to incorporate time is to explore the theory on manifolds with spatial boundaries. Over them some of the components of the metric are fixed, and a proper time over the spatial boundaries can be defined.

Here we focus on JT gravity [2, 3] over a strip with Lorentzian signature metrics. Our main motivation to choose a strip with Lorentzian metrics is to have an arena where time dependent non-perturbative QG amplitudes can be computed. In this work we will always refer to the Lorentzian time just by time.

As we are interested in computing QG amplitudes the strip should be bounded in time too, Fig. 1. Due to the four corners where the boundary is non-differentiable, studying QG on this manifold is harder than on its counterpart manifolds. Nonetheless, the non-smoothness of the corners can be handled by the inclusion of the Hayward term in the action [4, 5].

We compute the time dependent amplitude directly in the Lorentzian manifold in a non perturbative way. Although we do not include matter in this work, it can be regarded as a first step towards attacking the information paradox problem from a different perspective. Rather than computing entropies, we focus on the time dependent amplitudes to answer the question: is evolution in QG unitary? After all, entropies in QG could give some hints about unitarity, but a definitive answer about (non-)unitary evolution will come from the amplitudes and their behavior along time [1]. Our final result suggests that evolution is non-unitary for the strip with generic boundary conditions. However, there is a case where unitary evolution could be achieved. It is when when AdS$_2$, spatial boundary conditions are imposed [6–8].

Intriguingly, other wave functions related to other topologies obtained from the strip by gluing the time-like boundaries do not exhibit this non-unitary behavior. In fact, after tracing over the degrees of freedom defined on the spatial boundaries all references to the Lorenzian time disappears, as it should be for the QG wave function on the resulting geometries. This part of the work can also be seen as a complement to [9] and [10] where similar results have been found but using different methods to the ones presented here.

The paper is organized as follows. In the next section we define JT gravity on a Lorentzian strip. In section 3 we discuss generic boundary conditions on this manifold.

Although classically there could not exist solutions for generic boundary conditions, at the quantum level the story is different. Quantically, any boundary function is a good candidate to appear in the wave function. In this section, we also introduce the Lagrangian and the Hamiltonian form of the action. Special attention is paid to the Hayward terms, which are crucial [4, 5] to properly define the boundary action

and the degrees of freedom over the boundaries.

In section 4 we start the quantum exploration of JT gravity on the Lorentzian strip. We pose the problem for the transition amplitude we are interested in and show how to solve the gravitational path integral on this manifold. It is important to note that because we are assuming generic boundary conditions, in general there could not exists classical solutions for the metric. This means that integrating out first the dilaton field to solve the path integral is not an option here.

To solve the path integral we propose a canonical transformation inspired by the solution of the QG constraints [11, 12]. Although we do not use the conventional methods to solve a path integral, in the end we are able to fully find the amplitude because we alternate between the path integral and the canonical approach. Interestingly enough the time dependent amplitude indicates that the evolution is non-unitary for most of the boundary conditions.

In section 5 we show how to get other wave functions that are associated to other manifolds with different topologies. We get these manifolds after gluing the time-like boundaries of the strip. These wave functions do not reflect any non-unitary evolution. Conclusion are presented in section 6.

## 2 JT Action, Surface and Junction Terms

Let M, be a sufficiently well behaved 2-dimensional finite strip, where a time function t, which foliates M, into a set of constant t, space-like slices $\Sigma_t$, can be defined. The boundaries of M, consists of the initial and final space-like slices $\Sigma_i$, and $\Sigma_f$, see Fig. 1, as well as two time-like boundaries $B_L$, $B_R$. For every $\Sigma_t$. The intersections $\Sigma_i \cap B_{L,R}$, and $\Sigma_f \cap B_{L,R}$, are called the junctions and are denoted by $J_{i,}$, and $J_{f,}$, with an extra sub-index L, R depending on the location on the spatial boundaries. For the strip in two dimension the four junctions are just points.

The appropriate form of the Einstein Hilbert or JT action to make quantum gravity on manifolds with non-smooth boundaries Fig. 1 includes boundary, as well as, junction contributions. When we fix only the induced metric and the dilaton field on the boundaries the JT action reads as

$$S = \int_M d^2x \sqrt{-g}\, \phi(R - \lambda) - 2\sum_A \epsilon_A \int_A dy_A \sqrt{|h|}\, \phi K \tag{2.1}$$
$$+ 2\phi\eta \Big|_{J_{i,R}}^{J_{f,R}} - 2\phi\eta \Big|_{J_{i,L}}^{J_{f,L}}.$$

Where h, is fixed over $\Sigma_i$, $\Sigma_f$, $B_L$ and $B_R$. K, is the extrinsic curvature of the boundaries. The index A, runs over the boundaries; and $\epsilon_A$, is defined as

$$\epsilon_A = \begin{cases} +1, & \text{if } A = B_L \text{ , } B_R \\ -1, & \text{if } A = \Sigma_i \text{ , } \Sigma_f \text{ .} \end{cases}$$

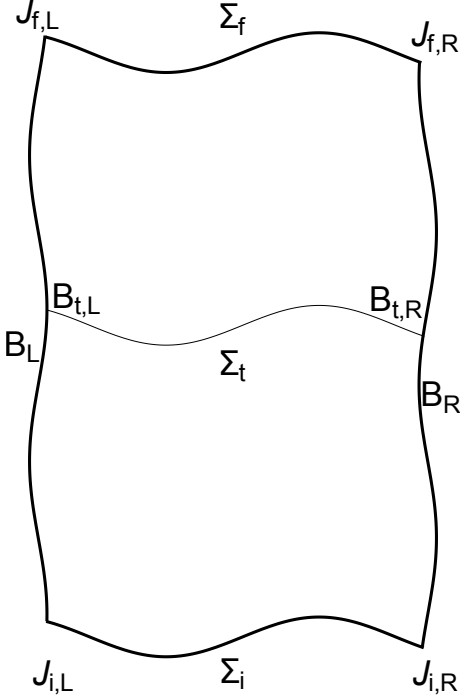

**Figure 1**. *The $(1+1)$ strip manifold M, the space-like boundaries $\Sigma_i$, and $\Sigma_f$, the time-like boundaries $B_L$ and $B_R$, and the junctions $J_{i,L}$, $J_{i,R}$, and $J_{f,L}$, $J_{f,R}$. $\Sigma_t$, a particular constant time slice and $B_{t,L}$ and $B_{t,R}$ the intersection $\Sigma_t \cap B_{L,R}$.*

The scalar function $\eta$, is defined as

$$\eta = \operatorname{arcsinh}\left(\hat{n}^{(\Sigma)}_{|B} \cdot \hat{n}^{(B)}\right), \tag{2.2}$$

where $\hat{n}^{(\Sigma)}_{|B}$, represents the two-dimensional unit normal to $\Sigma_t$, evaluated on $\Sigma_t \cap B$, and $\hat{n}^{(B)}$, is the two-dimensional unit normal to B. We would like to stress that $\eta$, is defined over the whole time-like boundaries $B_{L,R}$, including their boundaries, i.e., the junctions $J_i$, and $J_f$, and $\eta \in (-\infty, +\infty)$.

The variation of (2.1) is given by

$$\delta S = \int_M d^2x \sqrt{-g}\left[(R - \lambda)\delta\phi + \left(-\nabla_\mu\nabla_\nu\,\phi + \Box\,\phi\,g_{\mu\nu} + \frac{\lambda}{2}\phi\,g_{\mu\nu}\right)\delta g^{\mu\nu}\right] \tag{2.3}$$

$$- \sum_A \epsilon_A \int_{B_A} dy_A \sqrt{|h|}\left[\phi\left(K_{ab} - K\,h_{ab} + n^\sigma_{(A)}\partial_\sigma\,\phi\,h_{ab}\right)\delta h^{ab} + 2K\,\delta\phi\right].$$

After imposing $\delta h_{ab} = \delta\phi = 0$, over the boundaries, we get the vacuum JT's equations of motion. Note that no term including $\eta$, appears in (2.3).

# 3   Boundary Conditions, JT Lagrangian and Hamiltonian Actions and, Surface and Junctions Terms

In this section, we introduce the ADM decomposition of the two-metric. Without loss of the generality, we introduce a coordinate system over $\Sigma_t$, such that the spatial boundaries are located at a constant value of a coordinate that we call r. We perform a detailed analysis of the boundary conditions, to find the relation the lapse N, and the shift $N^r$, functions satisfy over the spatial boundaries. This relation evidences that while N, and $N^r$, are independent over M, over $B_{L,R}$, they are not.

In the Lagrangian formulation, this boundary relation between N, and $N^r$, is taken into account during the variation process of (2.1). However, we show that in the Hamiltonian formulation, this is not the case. Therefore, to take into account the variations of N, and $N^r$, over the spatial boundary it is convenient to introduce a new field to parameterize the boundary condition.

We present the Lagrangian and the Hamiltonian form of the JT gravity action in the ADM formalism and the boundary equation of motion associated with the new field. We discuss that while the variation of (2.1) does not lead to any boundary equation, the variation of the Hamiltonian action does lead to a boundary equation.

This boundary equation of motion is of algebraic type. This fact allows us to solve this equation in terms of the others boundary degrees of freedom and plug the solution back in the action. This treatment of the boundary action can also be found in [13, 14] but in a different context. In these two references, it can also be found a more detailed analysis of some of the issues discussed in this section but written in a different notation.

The ADM metric is given by

$$ds^2 = -N^2 dt^2 + \Lambda^2 (dr + N^r dt)^2. \tag{3.1}$$

On a constant time slice $\Sigma_t$, the induced metric takes the form

$$ds^2_{|_t} = \Lambda^2 dr^2, \tag{3.2}$$

and, $\sqrt{|h|} = \Lambda$. In particular on $\Sigma_{(i,f)}$, $ds^2_{|_{t_{(i,f)}}} = \Lambda^2_{(i,f)} dr^2$, with $\Lambda_{(i,f)}$, given functions of r. On the spatial boundaries the metric reads as

$$ds^2_{|_{r_{cons}}} = -\left(N^2 - (\Lambda N^r)^2\right) dt^2 = -\overline{N}^2 dt^2, \tag{3.3}$$

and, $\sqrt{|h|} = \overline{N}$.

From (3.3) we see that none of the degree of freedom N, $N^r$, or $\Lambda$, are fully specified over the spatial boundaries, instead what is fully specified is the combination

$$N^2 - (\Lambda N^r)^2 = \overline{N}^2, \tag{3.4}$$

Where $\overline{N}$, is a given function of $t$.

For the sake of completeness let us present the unit normal vectors to $\Sigma_t$, and $B_{L,R}$. Their expressions will be helpful for the subsequent discussions. In the coordinates $(t, r)$, one can define the functions

$$F^{(\Sigma)} = t_0 - t,$$
$$F^{(B)} = r_0 - r. \tag{3.5}$$

Such that a slice of constant time $t$, or constant r, are specified by the relations $F^{(\Sigma)} = F^{(B)} = 0$.

The unit normal vectors are defined as

$$\hat{n}_\mu^{(\Sigma)} = \frac{\partial_\mu F^{(\Sigma)}}{(-g^{\alpha\beta}\partial_\alpha F^{(\Sigma)}\partial_\beta F^{(\Sigma)})^{\frac{1}{2}}} = N(-1, 0), \tag{3.6}$$

$$\hat{n}_\mu^{(B)} = \frac{\partial_\mu F^{(B)}}{(g^{\alpha\beta}\partial_\alpha F^{(B)}\partial_\beta F^{(B)})^{\frac{1}{2}}} = \frac{(0, -1)}{(\Lambda^{-2} - N^{-2}N^r N^r)^{\frac{1}{2}}}.$$

The scalar product between both unit normal vectors over B, is

$$(g^{\mu\nu}\hat{n}_\mu^{(\Sigma)}\hat{n}_\nu^{(B)})_{|r_{cons}} = \frac{\Lambda N^r}{(N^2 - (\Lambda N^r)^2)^{\frac{1}{2}}} = \frac{\Lambda N^r}{\overline{N}}. \tag{3.7}$$

Now, (2.2) can be written as

$$\eta = \operatorname{arcsinh}\left(\frac{\Lambda N^r}{\overline{N}}\right). \tag{3.8}$$

Let us go back to discuss the boundary conditions expressed in the coordinates $(t, r)$. We would like to stress that we can not impose separately conditions on each functions appearing in (3.4).

Using (3.8) and (3.4) we note that

$$\Lambda N^r_{|r_{cons}} = \overline{N}\sinh(\eta), \tag{3.9}$$
$$N_{|r_{cons}} = \overline{N}\cosh(\eta).$$

Previous expressions indicate we can use $\eta$, to parameterize [1] the degrees of freedom over the spatial boundaries.

## 3.1 JT Lagrangian Action and Boundary Terms

In this section we introduce the Lagrangian ADM form of the JT gravity action, paying special attention to the boundary contribution. In two dimension the Ricci scalar can be written as

$$R = -2\nabla_\mu\left(K\hat{n}^\mu + \hat{n}^\nu\nabla_\nu\hat{n}^\mu\right), \tag{3.10}$$

---

[1]Although (3.4) can be parameterized in infinitely many ways, we find convenient to use (3.9) because it is the optimum one to carry out the subsequent calculations.

Using (3.10), from (2.1) we arrive at the JT Lagrangian form of the action,

$$S = 2 \int_M d^2x \left[ \Lambda(\partial_t\phi - N^r\partial_r\phi)K + \Lambda^{-1}\partial_r\phi\partial_r N - \frac{\lambda}{2}N\Lambda\phi \right] + 2 \int_{(B_R-B_L)} dt\, \partial_t\phi\, \eta, \quad (3.11)$$

where $K = -\Lambda^{-1}N^{-1}\left(\partial_t\Lambda - \partial_r(\Lambda N^r)\right)$, and

$$\int_{(B_R-B_L)} = \int_{B_R} - \int_{B_L}, \quad (3.12)$$

in the interval $[t_i, t_f]$. To get this action we have proceed in a similar way as in [15, 16], of course we have taken into account the particularities of 2d gravity. Note the unusual boundary term in (3.11). It is needed to have a well defined variation process and it will be essential in the Hamiltonian formulation. As $\phi$, is given over the boundaries, in particular $\partial_t\phi$, would be fixed over the spatial boundary too.

## 3.2   JT Hamiltonian Action and Boundary Terms

Let us introduce the JT Hamiltonian. From (3.11) we can compute the canonically conjugate momenta

$$P_\Lambda = -2N^{-1}\left(\partial_t\phi - N^r\partial_r\phi\right) = -2\hat{n}^\mu\partial_\mu\phi, \quad (3.13)$$
$$P_\phi = -2N^{-1}\left(\partial_t\Lambda - \partial_r(\Lambda N^r)\right) = -2\Lambda\, K.$$

After some manipulations, we arrive at

$$S = S_M + S_B, \quad (3.14)$$
$$= \int_{t_i}^{t_f} dt \int_{-r_0}^{r_0} dr \left[ P_\phi\partial_t\phi + P_\Lambda\partial_t\Lambda - N\mathcal{H} - N^r\mathcal{H}_r \right]$$
$$+ \int_{(B_R-B_L)} dt\left( \overline{N}\, P_\Lambda\, \sinh(\eta) - 2\overline{N}\Lambda^{-1}\partial_r\phi\, \cosh(\eta) + 2\partial_t\phi\, \eta \right),$$

where we have used (3.9) in the boundary action of (3.14), and

$$\mathcal{H} = -\frac{1}{2}P_\phi P_\Lambda + 2\partial_r\left(\Lambda^{-1}\partial_r\phi\right) + \lambda\, \Lambda\, \phi, \quad (3.15)$$
$$\mathcal{H}_r = \partial_r\phi P_\phi - \Lambda\partial_r P_\Lambda.$$

In the Hamiltonian formulation N, and $N^r$, play the role of Lagrange multipliers [17]. Their equations of motion only enforce the Hamiltonian and the momentum constraint $\mathcal{H} = \mathcal{H}_r = 0$, on M.

Variations of (3.14) with respect to N, N$^{\rm r}$, $\Lambda$, $\phi$, P$_\Lambda$, and P$_\phi$, over M, lead to an equivalent systems of equations of motion to that obtained from the Lagrangian (3.11). However, for the action written in the ADM Hamiltonian form extra care is needed with the variations over B$_{\rm L,R}$. Note that while the variations of the Lagrange multipliers N, and N$^{\rm r}$, are independent over M, over the boundaries B$_{\rm L,R}$, they are related each other through relation (3.4). Therefore, the variation of N, and N$^{\rm r}$, over B$_{\rm L,R}$, are not independent.

To take care of the variations of N, and N$^{\rm r}$, over B$_{\rm L,R}$, it is convenient to parameterize the solutions of (3.4), using (3.9), and regard $\eta$, as an independent degree of freedom defined over B$_{\rm L,R}$. This is the reason why the boundary action in (3.14) has been explicitly written in terms of $\eta$, and the others boundary degrees of freedom, instead of N, N$^{\rm r}$.

The boundary equations of motion, associated to $\eta$, on B$_{\rm L,R}$, read as

$$\overline{\rm N}\ {\rm P}_\Lambda\ \cosh(\eta) - 2\overline{\rm N}\Lambda^{-1}\partial_{\rm r}\phi\ \sinh(\eta) + 2\partial_t\phi = 0. \tag{3.16}$$

Equation (3.16) is contained in the first equation in (3.13). This can be checked by plugging (3.9) in (3.16). In other words, equation (3.16) can be viewed as the restriction to the time-like boundaries of the first equation in (3.13). Previous statement ensures there are no contradictions between the Lagrangian and the Hamiltonian formulations. Nonetheless, we emphasize that (3.16) and (3.13) come from the variation of two different and unrelated degrees of freedom. While the former comes from the variation of $\eta$, defined only over the spatial boundaries, the latter, in the Hamiltonian formulation, comes from the variation of P$_\Lambda$, over M, including its boundaries.

# 4   JT Quantum Gravity on the Strip

In this section we explore JT quantum gravity on the finite Lorentzian strip. Our final goal is to compute the time dependent transition amplitude. To this end we will use both, the canonical and the path integral formulation of gravity.

Before posing the problem we want to make an important remark. Note that integrating out first the field $\phi$, is not a clear option to solve the path integral in this manifold. We are not sure whether a classical solution satisfying the boundary conditions (3.2) and (3.3) for generic $\Lambda_{i,f}$, and for generic $\overline{\rm N}_{\rm L,R}$, does exist. For sure when $\Lambda_{i,f}$, and $\overline{\rm N}_{\rm L,R}$, are restricted to the cases of constant curvature spaces, the solution will exist and integrating out first the field $\phi$, becomes an option again.

This does not mean that we can not solve JT quantum gravity with generic boundary conditions. It means that the techniques used in the modern context of JT (modern in the sense that it is linked to SYK and Holography, with $AdS_2$ boundary conditions) are not applicable to this more general setup. It could be the case that for some boundary conditions there are not classical solutions but still there could be quantum transitions.

Let us now pose the problem. We are interested in computing the transition amplitude between an initial configuration $(\phi, \Lambda)_i$, at some initial time $t_i$, and a final one $(\phi, \Lambda)_f$, at some final time $t_f$, with spatial boundary values $(\phi, \overline{N})_R$ and $(\phi, \overline{N})_L$.

In the path integral formulation this transition amplitude is represented by

$$\Psi\Big[(\phi, \Lambda)_f, (\phi, \Lambda)_i \; ; (\phi, \overline{N})_R, (\phi, \overline{N})_L\Big] = \int D\big[N, N^r, \phi, \Lambda, P_\phi, P_\Lambda\big]\Big|_M$$
$$\int D\big[N, N^r, \Lambda, P_\Lambda\big]\Big|_{B_{L,R}}$$
$$\delta\big[N^2 - (\Lambda N^r)^2 - \overline{N}^2\big]\Big|_{B_{L,R}} e^{iS}. \qquad (4.1)$$

Conveniently we have split the measure into the bulk and the boundaries contribution. The functional Dirac delta enforces the boundary condition (3.4), over the time-like boundaries, and

$$S = \int\limits_{t_i}^{t_f} dt \int\limits_{-r_0}^{r_0} dr \Big[P_\phi \partial_t \phi + P_\Lambda \partial_t \Lambda - N\mathcal{H} - N^r \mathcal{H}_r\Big] \qquad (4.2)$$
$$+ \int\limits_{(B_R - B_L)} dt \Big(\Lambda N^r \, P_\Lambda - 2N\Lambda^{-1}\partial_r \phi + 2\partial_t \phi \, \text{arcsinh}\big(\frac{\Lambda N^r}{\overline{N}}\big)\Big).$$

We do not specified the measure of the path integral, because in the end the method developed here allows us to get the final form of the amplitudes with out performing an actual path integration.

After performing the change of variables over the spatial boundaries only

$$\Lambda N^r = \mathcal{R} \, \sinh(\eta), \qquad (4.3)$$
$$N = \mathcal{R} \, \cosh(\eta),$$

the transition amplitude can be written as

$$\Psi\Big[(\phi, \Lambda)_f, (\phi, \Lambda)_i \; ; (\phi, \overline{N})_R, (\phi, \overline{N})_L\Big] = \int D\big[N, N^r, \phi, \Lambda, P_\phi, P_\Lambda\big]\Big|_M$$
$$\int D\big[\eta, \Lambda, P_\Lambda\big]\Big|_{B_{L,R}} e^{iS}, \qquad (4.4)$$

where S, is as in (3.14). Given the form of the boundary action $S_B$, in principle we could point wise integrate $\eta$, to get Hankel or modified Bessel functions of the second kind. However, we do not know how to handle the infinity product of functions we get. Instead of integrating $\eta$, we will take advantage of the fact that its equation of motion is of algebraic type. In this case we are allowed to solve for $\eta$, in terms of the others boundary degrees of freedom and plug the solution in the action [13, 14]. We

stress that for algebraic equations of motion this is not a semi-classical procedure, so we still are in a non-perturbative regime. This procedure is completely equivalent to integrating in $\eta$.

The solution to (3.16), is

$$\eta = \ln\Big( \frac{(2\overline{N}^{-1}\partial_t\phi) + \sqrt{(2\Lambda^{-1}\partial_r\phi)^2 + (2\overline{N}^{-1}\partial_t\phi)^2 - P_\Lambda^2}}{(2\Lambda^{-1}\partial_r\phi) - P_\Lambda} \Big). \tag{4.5}$$

Plugging it in the boundary action we get

$$S_B = \int\limits_{(B_R - B_L)} dt\overline{N}\Big[ -\sqrt{(2\Lambda^{-1}\partial_r\phi)^2 + (2\overline{N}^{-1}\partial_t\phi)^2 - P_\Lambda^2} \tag{4.6}$$

$$+ (2\overline{N}^{-1}\partial_t\phi)\ln\Big( \frac{(2\overline{N}^{-1}\partial_t\phi) + \sqrt{(2\Lambda^{-1}\partial_r\phi)^2 + (2\overline{N}^{-1}\partial_t\phi)^2 - P_\Lambda^2}}{(2\Lambda^{-1}\partial_r\phi) - P_\Lambda} \Big)\Big].$$

Before moving to the bulk calculation we would like to make a further assumption which simplifies the next calculations. It will be implemented in two steps. First, we will assume that over the spatial boundaries $\phi$, is constant. This assumption translates into the conditions [6–8]

$$\partial_t\phi\Big|_{B_L,B_R} = 0. \tag{4.7}$$

Then, the transition amplitude (4.4) reduces to

$$\Psi\Big[(\phi,\Lambda)_f,(\phi,\Lambda)_i\ ;(\phi,\overline{N})_R,(\phi,\overline{N})_L\Big] = \int D\Big[N,N^r,\phi,\Lambda,P_\phi,P_\Lambda\Big]\Big|_M \tag{4.8}$$

$$\int D\Big[\Lambda,P_\Lambda\Big]\Big|_{B_{L,R}} e^{iS},$$

with S, given by

$$S = \int\limits_{t_i}^{t_f} dt \int\limits_{-r_0}^{r_0} dr\Big[ P_\phi\partial_t\phi + P_\Lambda\partial_t\Lambda - N\mathcal{H} - N^r\mathcal{H}_r\Big] \tag{4.9}$$

$$- \int\limits_{(B_R - B_L)} dt\ \overline{N}\sqrt{(2\Lambda^{-1}\partial_r\phi)^2 - P_\Lambda^2}.$$

Note that now, $\phi_R$, and $\phi_L$ are constants. In addition by consistency

$$\phi_i(-r_0) = \phi_f(-r_0) = \phi_R,$$
$$\phi_i(+r_0) = \phi_f(+r_0) = \phi_L. \tag{4.10}$$

### 4.1 Bulk Calculation

In order to show how to solve the full path integral (4.8) we find instructive first to look at the classical solution to the constraint $\mathcal{H} = \mathcal{H}_r = 0$, [11, 12].

This is

$$P_\Lambda = \left(C(t) + (2\Lambda^{-1}\partial_r\phi)^2 + 2\lambda\phi^2\right)^{\frac{1}{2}}, \tag{4.11}$$

$$P_\phi = 2\frac{\partial_r(2\Lambda^{-1}\partial_r\phi) + \lambda\Lambda\phi}{\left(C(t) + (2\Lambda^{-1}\partial_r\phi)^2 + 2\lambda\phi^2\right)^{\frac{1}{2}}},$$

where $C(t)$, is an arbitrary function of $t$. If we were performing this calculation over an space-time without spatial boundaries, for instance a cylinder, it would be enough to make the substitution $P_\Lambda = -i\frac{\delta}{\delta\Lambda}$, and $P_\phi = -i\frac{\delta}{\delta\phi}$, in (4.11) and solve for $\Psi$, the corresponding functional equations, i.e.,

$$-i\frac{\delta}{\delta\Lambda}\Psi = \left(C(t) + (2\Lambda^{-1}\partial_r\phi)^2 + 2\lambda\phi^2\right)^{\frac{1}{2}}\Psi, \tag{4.12}$$

$$-i\frac{\delta}{\delta\phi}\Psi = 2\frac{\partial_r(2\Lambda^{-1}\partial_r\phi) + \lambda\Lambda\phi}{\left(C(t) + (2\Lambda^{-1}\partial_r\phi)^2 + 2\lambda\phi^2\right)^{\frac{1}{2}}}\Psi.$$

In fact, these equations are equivalent to the Wheeler-DeWitt equation and the momentum constraint, with a particular factor ordering, see [12]. In this case we would find that

$$\Psi = \exp\left[i\,\Omega'[\Lambda, \phi; C]\right], \tag{4.13}$$

with

$$\Omega'[\Lambda, \phi; C] = \Omega[\Lambda, \phi; C] + G(C), \tag{4.14}$$

and

$$\Omega[\Lambda, \phi; C] = \int_{\Sigma_f} dr\, \Lambda\left[\sqrt{C + (2\Lambda^{-1}\partial_r\phi)^2 + 2\lambda\phi^2}\right. \tag{4.15}$$

$$\left. + (2\Lambda^{-1}\partial_r\phi)\ln\left(\frac{(2\Lambda^{-1}\partial_r\phi) + \sqrt{C + (2\Lambda^{-1}\partial_r\phi)^2 + 2\lambda\phi^2}}{(2\Lambda^{-1}\partial_r\phi) - \sqrt{C + (2\Lambda^{-1}\partial_r\phi)^2 + 2\lambda\phi^2}}\right)\right],$$

where $G(C)$, is an arbitrary function of the constant $C(t_f)$.

For the transition amplitude we are interested in, the former procedure does not apply as it stands because in the action (4.9) there is a boundary contribution that does not decouple from the bulk path integral.

The crucial observation here is that we can regard the solution (4.11) as an off-shell change of variables in the path integral. In fact it is a canonical transformation whose generating functional is given by $\Omega[\Lambda, \phi; C]$, (4.15), with no extra

boundary terms. A similar transformation has appeared in the literature before but in a different context within Einstein QG in four dimensions [18].

After performing the canonical transformation (4.11) to (4.9) we get (note that the transformation (4.11) maps $\mathcal{H}$, and $\mathcal{H}_r$, to zero)

$$
S = \int_{t_i}^{t_f} dt \int_{-r_0}^{r_0} dr \Big[ P_C \partial_t C + \tilde{P}_\phi \partial_t \phi - \mathcal{K}[C, \phi, P_C, \tilde{P}_\phi] \Big]
$$
$$
+ \int_{t_i}^{t_f} dt \frac{d}{dt} \Omega[\Lambda, \phi; C] - \int_{(B_R - B_L)} dt\, \overline{N} \sqrt{-C - 2\lambda\phi^2}.
$$

where

$$
P_\Lambda = \frac{\delta}{\delta\Lambda} \Omega[\Lambda, \phi; C], \tag{4.16}
$$
$$
P_\phi = \tilde{P}_\phi + \frac{\delta}{\delta\phi} \Omega[\Lambda, \phi; C],
$$
$$
P_C = \frac{\partial}{\partial C} \Omega[\Lambda, \phi; C],
$$

and the new Hamiltonian

$$
\mathcal{K}[C, \phi, P_C, \tilde{P}_\phi] = -\frac{1}{2} N \tilde{P}_\phi \sqrt{C + (2\Lambda^{-1}\partial_r\phi)^2 + 2\lambda\phi^2} + N^r \partial_r\phi \tilde{P}_\phi. \tag{4.17}
$$

In the new action the Lagrange multipliers impose the same constraint, $\tilde{P}_\phi = 0$. In the path integral this means that after integration in N the action will further reduce to

$$
S = \int_{t_i}^{t_f} dt\, \Pi_C\, \partial_t C + \Omega[\Lambda, \phi; C] \Big|_{t_i}^{t_f} - \int_{(B_R - B_L)} dt\, \overline{N} \sqrt{-C - 2\lambda\phi^2} \tag{4.18}
$$

where $\Pi_C(t) = \int_{-r_0}^{r_0} dr\, P_C(t)$, and the transition amplitude

$$
\Psi \Big[ (\phi, \Lambda)_f, (\phi, \Lambda)_i \,; (\phi, \overline{N})_R, (\phi, \overline{N})_L \Big] = e^{i\Omega[\Lambda, \phi; C] \big|_{t_i}^{t_f}} \int DC\, D\Pi_C
$$
$$
\exp\Big[ i \int_{t_i}^{t_f} dt\, \Pi_C\, \partial_t C + \tag{4.19}
$$
$$
- i \int_{(B_R - B_L)} dt\, \overline{N} \sqrt{-C - 2\lambda\phi^2} \Big],
$$

The new degree of freedom $C(t)$ is interpreted as a mass in the classical theory, this is why it is always greeter than zero. In the path integration it means that each

$C(t)$, with $t \in [t_i, t_f]$, will range from zero to infinite. Note that there is a value of $C(t)$, from which (4.18) become imaginary. On the one hand, If $\lambda > 0$, the action is imaginary for $C(t) \geq 0$. On the other hand, If $\lambda < 0$, the action is imaginary for $C(t) \geq 2|\lambda|\phi^2$. Certainly (4.11) is imaginary for most of the values of C.

To further proceed it is convenient to perform the $\Pi_C$, integral in (4.19). This integration gives rise a functional Dirac delta $\delta[\partial_t C]$. It imposes the condition C = const, while removes all the C integrals but one. In the end we get

$$\Psi\Big[(\phi, \Lambda)_f, (\phi, \Lambda)_i \; ; (\phi, \overline{N})_R, (\phi, \overline{N})_L\Big] = \int\limits_0^\infty dC \; \chi(C) \; e^{\; i\Omega[\Lambda, \phi; C]}\Big|_{t_i}^{t_f}$$

$$\exp\Big[\int\limits_{(B_R - B_L)} dt \; \overline{N}\sqrt{C + 2\lambda\phi^2}\Big]. \qquad (4.20)$$

Where $\chi(C)$, is coming from the fact that the new Hamiltonian $h[C, t]$,

$$h[C, t] = \big(\overline{N}_R(t) - \overline{N}_L(t)\big)\sqrt{-C - 2\lambda\phi^2}, \qquad (4.21)$$

in the new action is a function of C, [2]. This function $\chi(C)$, can be fixed by demanding initial conditions for the time dependent transition amplitude (4.20). Since C, and $\phi$, are constant we can easily introduce the invariant proper time $\tau$,

$$\tau = \int\limits_{t_i}^{t_f} dt \; \overline{N}. \qquad (4.24)$$

In this basis the transition amplitude reads

$$\Psi\Big[\phi, \Lambda_f, \Lambda_i \; ; \tau_R, \tau_L\Big] = \int\limits_0^\infty dC \; \chi(C) \; e^{\; i\Omega[\Lambda, \phi; C]}\Big|_{t_i}^{t_f}$$

$$\exp\Big[\sqrt{C + 2\lambda\phi^2}(\tau_R - \tau_L)\Big]. \qquad (4.25)$$

---

[2]To better understand the origin of $\chi(C)$, it is convenient to look at the quantum mechanical problem (4.19) à la Schrödinger. For this case the Schrödinger equation reads

$$i\partial_{t_f}\psi(C, t_f) = h[C, t_f] \; \psi(C, t_f) = \big(\overline{N}_R(t_f) - \overline{N}_L(t_f)\big)\sqrt{-C - 2\lambda\phi^2} \; \psi(C, t_f). \qquad (4.22)$$

It is straightforward to see that we can separate variables $\psi(C, t_f) = \chi(C)f(t_f)$, to finally get

$$\psi(C, t_f) = \chi(C)\exp\Big[\int\limits_{(B_R - B_L)} dt \; \overline{N}\sqrt{C + 2\lambda\phi^2}\Big]. \qquad (4.23)$$

When $t_f \to t_i$, $\tau = 0$, and the initial conditions are expressed as

$$\Psi\Big[\phi, \Lambda_f, \Lambda_i \; ; 0, 0\Big] = \int\limits_0^\infty dC \; \chi(C) \; e^{i\Omega[\Lambda, \phi; C]}\Big|_{t_i}^{t_f} = \Psi_0\Big[\phi, \Lambda_f, \Lambda_i\Big], \qquad (4.26)$$

with $\Psi_0\Big[\phi, \Lambda_f, \Lambda_i\Big]$, a given functional representing the initial state. Note that when $t_f \to t_i$, not necessarily $\Lambda_f \to \Lambda_i$. Solving the functional equation (4.26) for $\chi(C)$, complete determines the time dependent state.

At his point some comments are in order. Relation (4.26) determines the function $\chi(C)$, as in ordinary quantum mechanics a similar expression determines the Fourier coefficients of a given general solution of the Schrödinger equation. In ordinary quantum mechanics to fully specify a state at some time $t > 0$, one needs to prescribed the state at $t = 0$.

Relation (4.26) can be seen as a generalization of the statements in previous paragraph. note that it is a functional relation and the function $\chi(C)$ plays the role of the Fourier coefficients. Note that the right hand side is a given functional, namely the prescribed initial state at $\tau_R = \tau_L = 0$. In other words, as in ordinary quantum mechanics, here one needs to prescribed the state at some initial time in order to fully determine the time dependent amplitude we are interested in.

Now, we implement the second part of the assumption we implemented in previous section. Assuming that $\phi$, is constant over the space-like boundaries implies

$$\partial_r \phi\Big|_{\Sigma_i, \Sigma_f} = 0. \qquad (4.27)$$

Then, introducing the invariant proper length l,

$$l = \int\limits_{-r_0}^{r_0} dr \; \Lambda, \qquad (4.28)$$

we have

$$\Psi\Big[\phi, l_f, l_i \; ; \tau_R, \tau_L\Big] = \int\limits_0^\infty dC \; \chi(C) \exp\Big[\sqrt{C + 2\lambda\phi^2}\Big(i(l_f - l_i) + (\tau_R - \tau_L)\Big)\Big], \quad (4.29)$$

Note, that we can rewrite the transition amplitude as

$$\Psi\Big[\phi, l_f, l_i \; ; \tau_R, \tau_L\Big] = \int\limits_0^\infty dC \; \chi(C) \exp\Big[\sqrt{C + 2\lambda\phi^2} \; i \; L\Big], \qquad (4.30)$$

where $L = \oint ds$, is taken clockwise around the boundary

$$L = \int\limits_{r_0}^{-r_0} ds\Big|_{\Sigma_i} + \int\limits_{t_i}^{t_f} ds\Big|_{B_L} + \int\limits_{-r_0}^{r_0} ds\Big|_{\Sigma_f} + \int\limits_{t_f}^{t_i} ds\Big|_{B_R}. \qquad (4.31)$$

From (4.24) it is straightforward to see that (4.29) is invariant under the corner-preserving reparametrizations. Additionally we can see that (4.30) satisfies a sort of reduced Wheeler-DeWitt equation with complex variable $L = (l_f - l_i) - i(\tau_R - \tau_L)$,

$$\left(\partial_L (L^{-1} \partial_\phi) + 2\lambda\phi\right) \Psi[\phi, L] = 0. \tag{4.32}$$

We stress that we are not developing a mini-superspace model of JT gravity. The appearance of (4.32), which is similar to the equation one gets in some mini-superspace models, see for instance [9] (for $L \in \mathbb{R}$), is a consequence of the restriction of the dilaton field to constant values over the boundaries. Nonetheless (4.32), could be helpful for determining some of the amplitudes.

We would like to emphasize that to fully determine $\Psi\left[\phi, l_f, l_i, \tau_R, \tau_L\right]$, we need to specify $\chi(C)$, which for the amplitudes we have computed it is not interpreted as a density of states. In general, and unlike the calculation of the partition function, it could take negative values, or it could even be an imaginary function. Moreover, it does not need to be derived from something else. It is just fixed by demanding initial conditions (4.26) for the amplitude. It is worth to remark however that finding $\chi(C)$, from (4.26), in general, could be involved without knowing explicitly the product between states [3].

Let us now study the behavior in time of (4.25). Note that in (4.25) we can have a varying $\phi$, and $\Lambda$, over $\Sigma_i$, and $\Sigma_f$. Note also that (4.25) contains a growing (decreasing) time dependent exponential factor. This is an indication that, for some states the probability will not be conserved.

The first conclusion one can make is that for $\lambda \geq 0$, (4.25) will evolve non-unitary. On the other hand, if $\lambda < 0$, there are states that potentially could evolve unitary. For instance, for the particular state preparation [4]

$$\chi(C) = \sum_a \delta(C - C_a). \tag{4.34}$$

If $C_a - 2|\lambda|\phi^2 < 0$, for all $C_a$, clearly the probability is conserved. Nonetheless these states describe unrealistic scenarios.

To have more realistic scenarios we need to have $C - 2|\lambda|\phi^2 < 0$, for all values of $C \in [0, \infty]$. Note that this happens only when $\phi_{|_{B_{(R,L)}}} \to \infty$, or

$$\phi_{|_{B_{(R,L)}}} = \frac{1}{\epsilon}, \tag{4.35}$$

---

[3]If there exists a functional measure $M[\phi_f, \Lambda_f]$, such that

$$\int D\phi_f D\Lambda_f \, M[\phi_f, \Lambda_f] \, \exp\left[i\left(\Omega[\Lambda_{t_f}, \phi_{t_f}; C] - \Omega[\Lambda_{t_f}, \phi_{t_f}; C']\right)\right] = f(C)\delta(C - C'), \tag{4.33}$$

The product (4.33) can be considered as the inner product in the Hilbert space.

[4]Choosing $\chi(C)$, could be considered equivalent to prescribe the initial state $\Psi_0$, in (4.26), see also at the end of section (5).

with $\epsilon \to 0$. These conditions can be recognized with the $AdS_2$, boundary conditions [6–8]. In this case we can take the metric near the spatial boundaries, located at $\sigma = 0$, and $\sigma = \pi$, to be

$$ds^2 = \frac{-dt^2 + d\sigma^2}{\sin^2(\sigma)}. \tag{4.36}$$

For this metric the boundary condition for $\overline{N}$, rewrites as

$$\overline{N}_{(R,L)} = \frac{\overline{N}_{0\ (R,L)}}{\epsilon}. \tag{4.37}$$

Recall that we can not regard $\phi$, as a constant over the four boundaries. In fact now we need to have a varying dilaton over the initial and the final slices, as well as a varying $\Lambda$. Note that if the dilaton were constant over the initial and the final slices then by continuity of the function (4.10), it would be infinity over them. The issue is that with a constant dilaton over $\Sigma_i$, and $\Sigma_f$, the limit $\epsilon \to 0$, of the action in (4.8) is not well defined. In addition there not exists counterterms that can fix this limit. Consistency among boundaries conditions (4.10) implies that

$$\lim_{\sigma \to B} \phi_{(i,f)} = \infty, \tag{4.38}$$

for the varying dilaton over $\Sigma_i$, and $\Sigma_f$.

Starting from (4.20), it is straightforward to see that

$$\Psi\Big[(\phi, \Lambda)_f, (\phi, \Lambda)_i\ ; (\infty, \infty)_R, (\infty, \infty)_L\Big] = \int\limits_0^\infty dC\ \chi(C)\ e^{i\Omega[\Lambda,\phi;C]}\Big|_{t_i}^{t_f}$$
$$\exp\Big[-i\frac{C}{2\sqrt{2\lambda}}(\tau_R - \tau_L)\Big]. \tag{4.39}$$

Where we have removed the infinity contribution $\frac{\sqrt{2\lambda}}{\epsilon^2}(\tau_R - \tau_L)$. From this term one can read of the counterterm needed in the action (2.1) to make it well defined for these particular boundary conditions. It is given by

$$S_{\text{counterterm}} = -\sqrt{2\lambda} \int\limits_{(B_R - B_L)} \phi\ ds. \tag{4.40}$$

Now it is clear to see that with the proper product between states (4.33) and, because of the appearance of the complex unit in front of the proper time in the exponent of (4.39) this amplitude might evolve unitary [5].

---

[5] If (4.33) holds, the probability does not dependent on time and it is given by

$$P(\tau) = \int\limits_0^\infty dC\ \chi(C)\chi^*(C)\ f(C) = P(0). \tag{4.41}$$

# 5 Other topologies

In what follows we will see how by gluing the two spatial boundaries we can make contact with some known results. To glue these two boundaries we can not consider constant functions over the spatial boundaries.

Starting from (4.2) or (4.6) we immediately see that once we identify the functions defined over $B_R$, with the ones over $B_L$, their contribution cancel each other. So, after tracing over the boundary degrees of freedom we will get a multiplicative infinity constant that can be ignored.

From this point on we can proceed similarly to previous section to get

$$\Psi\Big[(\phi,\Lambda)_f,(\phi,\Lambda)_i\Big] = \int\limits_0^\infty dC\ \chi(C)\ e^{\ i\Omega[\Lambda,\phi;C]}\Big|_{t_i}^{t_f}, \tag{5.1}$$

defined over a cylinder. Now for simplicity we can restrict the boundaries functions to be constants. In this case we have

$$\Psi\Big[(\phi,l)_f,(\phi,l)_i\Big] = \int\limits_0^\infty dC\ \chi(C)\exp\Big[i\sqrt{C+2\lambda\phi_f^2}\ l_f - i\sqrt{C+2\lambda\phi_i^2}\ l_i\Big]. \tag{5.2}$$

In principle there should exists a function $\chi(C)$, such that from (5.2) we get the finite cutoff Lorentzian version of the cylinder propagator, see appendix A. However, here we have not been able to fully find this function, see [9] for a discussion about this topic.

In addition, when we set to zero the length of the initial segment/circle, depending at which stage of the calculation we shrink it, the strip/cylinder might reduce to a disk with a conical defect [9, 19] or a cap. Now, we can get the corresponding amplitudes from (5.2). After setting $l_i = 0$, we can see that we do not need to specify $\phi_i$, any more. As it should be for these geometries. In the end we have

$$\Psi[\phi,l] = \int\limits_0^\infty dC\ \chi(C)\exp\Big[i\sqrt{C+2\lambda\phi^2}\ l\Big]. \tag{5.3}$$

To make contact with known results, or contrast with them, we will impose the so-called *no-boundary condition*, as a boundary condition in the space of wave function $\Psi$. This boundary condition fixed the function $\chi(C)$ to the very well known form, see [9], and reference there in.

$$\chi(C) = \sinh\big(2\pi\sqrt{C}\big). \tag{5.4}$$

We would like to remark that the appearance of (5.4) it just a consequence of the boundary condition we want to impose in the space of wave functions. Clearly if we

demand a different boundary conditions in the space of wave functions we would get a different $\chi(C)$, hence a different state. Integrating (5.4) with $\lambda > 0$, we can make contact with the $dS_2$ Lorentzian version of the finite cutoff wave function.

$$\Psi[\phi, l] = \frac{\phi^2 \, l}{(l^2 - 4\pi^2)} H_2^{(1)}\left(\sqrt{2\lambda} \, |\phi|(l^2 - 4\pi^2)^{\frac{1}{2}}\right), \tag{5.5}$$

where $H_2^{(1)}$, is the Hankel function of first order. This wave function have been found in [9] in the context of finite cutoff JT gravity for the contracting branch of the wave function. Our result differs a bit from them because we are considering the expanding branch [6].

Although (5.3) seems problematic when $\lambda < 0$, the analytic extension of (5.5) to these values of $\lambda$ is well defined. In this case we get what could be interpreted as the Lorentzian version of the finite cutoff JT wave function for $AdS_2$, and *no-boundary condition*, as boundary condition for $\Psi$.

$$\Psi[\phi, l] = \frac{\phi^2 \, l}{(l^2 - 4\pi^2)} K_2\left(\sqrt{2|\lambda|} \, |\phi|(l^2 - 4\pi^2)^{\frac{1}{2}}\right), \tag{5.6}$$

where $K_2$, is the modified Bessel function of the second kind.

Now, we are in a condition of constructing more explicitly what could be a state that behave non unitary in time in the Lorentzian strip. We would like to study (4.29) on the strip for the choice (5.4). The analytically extended (to all values of $\tau_R - \tau_L$ ) new function with $\lambda > 0$, is given by (5.5) where $l \to L = (l_f - l_i) - i(\tau_R - \tau_L)$. This function is regular for all values of $(\phi, l_f, l_i, \tau_R, \tau_L)$, except when both time-like boundaries have the same boundary condition, $\overline{N}_R - \overline{N}_L = 0$ (or $\tau_R - \tau_L = 0$), and $|l_f - l_i| = 2\pi$.

Starting from (4.29), we get

$$\Psi\left[\phi, l_f, l_i \; ; \tau_R, \tau_L\right] = \frac{\phi^2 \, L}{(L^2 - 4\pi^2)} H_2^{(1)}\left(\sqrt{2\lambda} \, |\phi|(L^2 - 4\pi^2)^{\frac{1}{2}}\right). \tag{5.7}$$

Note that for the Lorentzian strip, with the choice (5.4) effectively what we are imposing is the initial condition.

$$\Psi\left[\phi, l_f, l_i \; ; 0, 0\right] = \frac{\phi^2 \, l}{(l^2 - 4\pi^2)} H_2^{(1)}\left(\sqrt{2\lambda} \, |\phi|(l^2 - 4\pi^2)^{\frac{1}{2}}\right) = \Psi_0[\phi . l_f, l_i]. \tag{5.8}$$

In this way we can construct many more states evolving potentially in a non-unitary way.

---

[6] We have not used the term expanding (contracting ) branch over the paper. Nonetheless, (4.13) can be recognize with the expanding branch. Te contracting branch is defined with opposite sign in the exponent.

# 6  Conclusions

In this work we have studied JT quantum gravity on a finite strip with Lorentzian metrics. This manifold provides a natural setting for computing time dependent amplitudes.

We have noted that independent of the boundary condition $\overline{\mathrm{N}}$, (3.4) the three degrees of freedom $(\mathrm{N}, \mathrm{N^r}, \Lambda)$, that are independent in the bulk, get entangled at the time-like boundaries. In the Lagrangian formulation relation (3.4) is taken into account during the variation process. In the Hamiltonian formulation, where $(\mathrm{N}, \mathrm{N^r})$, play the role of Lagrange multipliers (3.14), the variation with respect to these fields must be taken carefully.

Although one could consider $(\mathrm{N}, \mathrm{N^r})$, as the degrees of freedom all the way until the spatial boundaries, we have shown that it is more convenient to use the Hayward angle $\eta$, (2.2) to parameterize $(\mathrm{N}, \mathrm{N^r})$, (3.9) over the time-like boundaries. In this way, varying $\eta$, is equivalent to varying $(\mathrm{N}, \mathrm{N^r})$ over the spatial boundaries while the condition (3.4) is respected.

The equation of motion associated to $\eta$, (3.16) is the restriction to the time-like boundaries of the first equation in (3.13). This is a desired fact because it indicates that there are not contradiction between the Lagrangian and the Hamiltonian formulations. In the former one never encounters a boundary equation of motion, see section 2. It is important to note, however, that from the action in the Hamiltonian form (3.14), these two equations come from two different and independent degrees of freedom. One comes from the variation of $\eta$, while the other comes from the variation of $\mathrm{P}_\Lambda$.

In section 4 we start the exploration of JT quantum gravity. We pose the gravitational path integral for the amplitude we are interested in (4.1), and solve it in two parts.

In the first part we focus on the time-like boundaries degrees of freedom, specifically $\eta$. Although we know how to point wise solve the integrals for $\eta$, leading to Hankel or modified Bessel functions of the second kind [7], we do not know how to handle the infinite product we get. Certainly it would be nice if we could compare the result of this infinite product with the result obtained here by just following the observation that the equation for $\eta$, is of algebraic type.

From this observation, instead of solving the integrals we solve the classical equation of motion associated to $\eta$, (3.16) in terms of the other boundary degrees of freedom and plug the solution back into the action. It might look here we are proceeding in a semi-classical way but it is not the case. We still are performing this calculation in a non-perturbative fashion. This is an standard procedure for this

---

[7]From (4.4) and (3.14) we can see that the $\eta$, integrals are the integral representation of the Hankel functions.

kind of actions. For instance, in string theory thanks to this procedure we can go from the Polyakov to the Nambu-Goto action.

In the second part we focus on the bulk degrees of freedom. Here to fully find the amplitude we alternate between the path integral and the canonical approach. As it is clear at this stage, we can not integrate the dilaton field in order to solve the path integral. Instead, and based on the classical and the quantum solutions of the constraints (4.11)-(4.15), we propose a canonical transformation (4.16) whose generating functional is given in (4.15). After some simplifications, we finally arrive at the time dependent QG amplitude (4.30). Surprisingly, (4.30) indicates that the evolution could be non-unitary for most of the boundary conditions. More surprising perhaps is the fact that for the particular case of $AdS_2$ boundary conditions unitarity is recovered (4.39).

As a complement and to make contact with some results found in [9] and [10], or at least contrast with some of the results in these references, we present the section 5. Here we explore the cylinder and the cap manifolds as spaces obtained from gluing the time-like boundaries of the strip. We find the propagator on the cylinder (A.2) and the $dS_2$, and $AdS_2$, Lorentzian versions of the cap wave functions (5.5) and (5.6). Remarkably, in these wave functions there are not references to any time thus there are not indications that evolution could be non-unitary. In the end we discuss a clear example of non-unitary evolution on the strip for the particular choice (5.4) of the function $\chi(C)$.

As explained in [9] finite cutoff JT gravity with $AdS_2$, boundary conditions can be interpreted as a $T\overline{T}$, deformation of the Schwarzian action. This was checked in Euclidean JT gravity. Applying similar reasoning here we should expect that an holographic interpretation for what we have done here could be given in terms of a $T\overline{T}$ deformed Schwarzian theory with Lorentzian signature. This connection will be explored elsewhere.

## Acknowledgments

We are grateful to Jose J. Fernandez-Melgarejo and Javier Molina-Vilaplana for comments and support. We would also like to thanks Pablo Bueno, Roberto Emparan and Arpan Bhattacharyya for discussions, comments and suggestions and to the member of the gravitation group of ICREA at the Barcelona University for the hospitality and the interesting discussion during the presentation of this work.

This work is supported by Next Generation EU through the Maria Zambrano grant from the Spanish Ministry of Universities under the Plan de Recuperación, Transformación y Resiliencia, and by the Spanish Ministerio de Innovación y Ciencia, CARM Fundación Séneca and Universidad de Murcia under the grant PID2021-125700NA-C22, 21257/PI/19.

# A    Cylinder propagator

In a pure speculative sense we would like to make some comments on the cylinder propagator. From a third quantization point of view this object satisfies the equation [9]

$$\left(\partial_{l_f}(l_f^{-1}\partial_\phi) + 2\lambda\phi\right)\Psi\Big[(\phi,l)_f,(\phi,l)_i\Big] = \frac{\delta(l_f - l_i)\delta(\phi_f - \phi_i)}{l_f}. \tag{A.1}$$

We have noted that for the choice $\chi(C) \propto \frac{d}{dC}(\frac{1}{\sqrt{C}})$, we can get this propagator from (5.2)

$$\Psi\Big[(\phi,l)_f,(\phi,l)_i\Big] \propto l_f H_0^{(1)}\left(\sqrt{2\lambda}\sqrt{(l_f^2 - l_i^2)(\phi_f^2 - \phi_i^2)}\right), \tag{A.2}$$

in the cases $(l_i = 0, \phi_i \neq 0)$ and $(l_i \neq 0,\ \phi_i = 0)$. To solve this integral for these cases we first regularize the integral i.e., $\int_0^\infty \rightarrow \int_\epsilon^\infty$. Then after integration by parts, with the substitution $C = 2\lambda\phi^2\sinh^2(s)$, with $\lambda > 0$ the integral can be straightforwardly solved. As a final step we remove all the divergent terms and take the limit $\epsilon \rightarrow 0$.

Note that with the change of variables

$$x = l^2 + \phi^2,$$
$$y = l^2 - \phi^2,$$

We can bring the Helmholtz equation (A.1) into an equation in Minkowski space with a Dirac delta source. The solution to this equation is the Green function

$$H_0^{(1)}\left(\sqrt{2\lambda}\sqrt{(x - x')^2 - (y - y')^2}\right). \tag{A.3}$$

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
