# Peer review of "JT Gravity on a Finite Lorentzian Strip: Time dependent Quantum Gravity Amplitudes"

_SciPost Physics_

## Round 2 · Referee Report · Anonymous (Referee 1) · 2023-6-9

Strengths

1- Considers JT gravity in the presence of spacelike plus timeline boundaries plus corners, which is more general than most existing analyses

2- Discusses quantum aspects, including arbitrary topologies

Weaknesses

1- Pertinent work missing from the references has substantial overlap with the first half of the paper (see report)

Report

The paper deals with (Lorentzian) JT gravity on a finite strip, i.e., on a spacetime region bounded by four corners, which in turn are connected by spacelike and timeline boundaries. This is nicely captured in Fig. 1.

Note, however, that Fig. 1 is essentially identical to Fig. 1 in https://arxiv.org/abs/gr-qc/9612021, which is not cited. Moreover, that reference has substantial overlap with the first half of the paper (albeit in somewhat different notation). In particular, the setup on a finite Lorentzian strip, the relevance of boundary and corner terms, the introduction of the rapidity variable, the imposition of suitable boundary conditions, and related issues were all done already in that paper. At least, I was not able to find anything new in the first three sections as compared to the paper from '96.

I am a bit unsure whether or not section 4 contains new content, though it might. Section 5 seems to be genuinely new, combining the results of the '96 paper with more recent discussions along the lines of Ref. [6].

Overall, the results in sections 4-5 (together with the ingredients from sections 2-4) seem sufficiently interesting so that eventually the paper could be published in SciPost Physics.

However, some changes are required (see requested changes).

Requested changes

1- My main recommendation to the author is to carefully read the '96 paper quoted above and to check (and state clearly) which of the statements until the end of section 4 are already covered by that work. While I do not necessarily think the paper has to be shortened (since there is value in it being self-contained), it should be pointed out clearly which of the results are novel and which are contained already in the '96 paper.

2- Given that the '96 paper applies to models more general than JT it seems natural to pose the question whether or not the results of sections 4-5 might be generalized beyond JT as well. Perhaps the author can comment on such generalizations in the conclusions.

3- Sticking with JT gravity, it is natural to ask about the holographic interpretation of the results. Is there some clear statement of what the authors' construction means in the context of the SYK model? If so, this should be mentioned in the conclusions. If not, it might still be worthwhile to point toward such considerations in the conclusions.

  • validity: high
  • significance: high
  • originality: ok
  • clarity: high
  • formatting: excellent
  • grammar: excellent

Author:  Jose Alejandro Rosabal Rodriguez  on 2023-06-16  [id 3736]

(in reply to Report 1 on 2023-06-09)
Category:
answer to question

Dear referee

Thank you so much for this fair report. I agree with you in all points and these issues will be addressed in the new version of the manuscript.

1- The paper you mention will be properly cited. I would like to make some comments on this point because it is true that this 96 paper has some overlap with the first part of my paper and I was not aware of this work. It is worth mentioning that I got these results independently in a more general context reported in ref. 11. Also that the notation in this 96 paper is quite involved and it discusses topics that at some point overlap with mine but  they are orthogonal to my work. Nonetheless, in general, this work will be a good complement for mine.

2-Regarding the other results  I used, first,  they are properly cited in my work. Second, I would like to add that unfortunately nowadays a huge amount of calculations has been performed in different contexts, so it is quite likely that some calculations today use similar methods to those used in the past. What we should highlight here, as you already mentioned,  is that overall, combining all these old methods and results with my new ideas I get something completely new.  

3- Although some results in section 1-3 are presented in  the 96 paper, in my work they are presented  clearly and combined in a novel way that is adapted for the quantum calculations in section 4 and 5. Making the paper self contained.

The subsequent results in section 4 and 5 are new. Moreover, the canonical transformation in my work is motivated by the classical solution of the constraints. In this way we establish a connection between the early work in ref. 8 and ref. 13 (or what would be the 2d version of it). This is also a straightforward method to find any generating function associated with any possible solution of the constraints equ's. (4.11) - (4.14) (possible with matter). This procedure could potentially  be exploited to solve similar problems in higher dimensions. For instance the calculation of the time dependent amplitude in a spherically symmetric quantum gravity.

4- The requested changes will appear in the new version.

---

## Round 2 · Referee Report · Chitraang Murdia (Referee 2) · 2023-9-24

Strengths

1. This paper aims to further the understanding of Lorentzian quantum gravity. The computation of the transition amplitude is very interesting.

2. The paper is self-contained and detailed.

Weaknesses

1. Although the paper tries to make contact with known results about the double trumpet wave function, it is not clear if this relationship exists.

Report

Understanding Lorezntian quantum gravity is a very important goal in the field. This paper attempts to further this goal by studying Lorentzian JT on a finite strip.

The results are novel and interesting. I would recommend that this paper be published after minor revision.

Requested changes

1. The function χ(C) is an important object in the analysis. It would be great to have some comments made on what this function means physically. Also, the connection to unitarity can be explored further.

2. It would be nice to include some comments on how to incorporate topology change (e.g., splitting/merging of baby universes) in this framework.

---

## Round 3 · Referee Report · Anonymous (Referee 3) · 2023-11-9

Strengths

1. Overall, the philosophy of the paper is interesting with the potential to yield a toy model for amplitudes in quantum gravity

Weaknesses

1. Conclusions are overstated and not supported by his calculations: it is unclear whether the evolution is non-unitary.

2. The section about other topologies should be clarified.

3. Unclear how to match other JT results in the limit $\tau_{1,2}$, $l_{1,2}$, $\phi \to \infty$ which is necessary to check the validity of the computations.

Report

The paper under review attempts to compute the transition amplitude between two different states in JT gravity along an AdS$_2$ Lorentzian strip, for which the sizes of the initial and final spatial slices, as well as the two boundary times, can all be finite. The paper uses a mixture of canonical quantization and path integral methods to compute this amplitude. While the methods and overall philosophy of this paper are interesting and I highly encourage the author to further explore the properties of this amplitude, I do not believe that the paper yet meets the quality standard of SciPost. In particular, I believe that the author’s conclusions are overstated and not supported by his calculations.

My main concern is that in order to be able to support the idea that this toy model exhibits non-unitary evolution, the author should present a convincing way to determine the function $\chi(C)$ that is undetermined by the Wheeler-de Witt equation. Can $\chi(C)$ be determined by matching the amplitude to a quantity computed in JT gravity solely from the path integral? Perhaps this can be done in the limit $\tau_{1,2}$, $l_{1,2}$, $\phi \to \infty$ and taking an analytic continuation to Euclidean signature. Should the resulting amplitude match the partition function in JT gravity since the boundary conditions along the four segments become the same as those for an asymptotic AdS$_2 $ boundary? Can that be used to determine$ \chi(C)$?

More broadly, it is unclear whether the transition amplitude in gravity along a finite strip should be a unitary process. In particular, if one considers the partition function of 2d gravity on a manifold whose boundary has a finite proper length, the density of states obtained by the Laplace transform has (at least naively) support on states that have a non-zero imaginary part. As discussed in reference [6] in the paper under review, this is related to the choice of the function that is undetermined by the WdW equation $\chi(C)$.

Additionally, I believe the ideas presented in section 5 should be clarified. Is the author claiming that the results found, for instance, in (5.3), should be viewed as corrections to the presumably leading transition amplitude found in section 4? The two geometries satisfy different boundary conditions (the one in section 4 has timelike boundaries while the geometry studied in section 5 does not), so that should not be the case. It is indeed the case that there are geometries with other topologies that satisfy the same boundary conditions that contribute to the transition amplitude. These could be analytic continuations of the handle disk or of other higher-genus geometries; nevertheless, the author does not compute their contribution.
Moreover, it is unclear whether the calculation (5.3) captures the contribution of a hyperbolic cylinder or a hyperbolic annulus. Namely, does the geometry described by (5.3) have a closed geodesic as on the cylinder, or does it not as on the annulus? Is that again determined by the choice of $\chi(C)$?

  • validity: ok
  • significance: good
  • originality: high
  • clarity: good
  • formatting: excellent
  • grammar: excellent

Author:  Jose Alejandro Rosabal Rodriguez  on 2023-11-16  [id 4119]

(in reply to Report 1 on 2023-11-09)
Category:
answer to question

Dear referee

Thank you very much for giving me the opportunity of answering this report. Certainly, responding it will strength my work and will clarify some conceptual issues I have found in the report.

[The paper under review attempts to compute the transition amplitude between two different states in JT gravity along an AdS2. Lorentzian strip, for which the sizes of the initial and final spatial slices, as well as the two boundary times, can all be finite. ]

This description of my work is inaccurate. In the paper I compute the amplitude, or wave function, for an space that is topologically a finite strip (polygon with 4 sides). The metrics on this manifold are considered to have Lorentzian signature.

More importantly, the boundary value of the metric and the dilaton field are GENERIC. This means that I do not exclusively discuss the case of AdS_2 boundary conditions, nor I put JT gravity on an AdS_2 space.

Perhaps in the abstract and the conclusions I should have emphasized more that all calculations are performed for GENERIC boundary conditions.

For instance, the only reference in the abstract relate to the fact I use generic boundary conditions is in this sentence:

The final result suggests that time evolution is non-unitary for most of the boundary conditions.

Certainly, this point should be clarified and expanded in the abstract and the conclusions. Nonetheless, I want to emphasize here that in the main text it is clear that the calculations are performed for GENERIC boundary conditions. For instance in the introduction you can find the first reference to that in the fourth paragraph in the sentence:

Our final result suggests that evolution is non-unitary for the strip with generic boundary conditions

Followed of six more references along the paper, and several calculations pointing to the this fact. One the most important is given in page 3:

It is important to note that because we are assuming generic boundary conditions, in general there
could not exists classical solutions for the metric. This means that integrating out
first the dilaton field to solve the path integral is not an option here.

Similar comment has been repeated at the beginning of section 4.

[The paper uses a mixture of canonical quantization and path integral methods to compute this amplitude. While the methods and overall philosophy of this paper are interesting and I highly encourage the author to further explore the properties of this amplitude,]

This is correct and thank you very much for encouraging me to continue on this line. In fact this is what I did. You can check the work 2309.03639 where I extend in a novel way the results of the paper under review to the spherically symmetric space time in four dimensions.

[I do not believe that the paper yet meets the quality standard of SciPost. In particular, I believe that the author’s conclusions are overstated and not supported by his calculations.]

All the calculations supporting the conclusions appear in the paper and this can easily checked just by looking at how many equations have been referenced in the conclusion.

Having pointed out these deficiencies in the report (and a few more I am going to) I consider that the referee´s complain about the quality of the paper, and its conclusions is based on a wrong appreciation of my results. Nonetheless, I agree that some minor clarifications are needed to emphasize that calculations are performed with generic boundary conditions.

[My main concern is that in order to be able to support the idea that this toy model exhibits non-unitary evolution, the author should present a convincing way to determine the function χ(C) that is undetermined by the Wheeler-de Witt equation.]

The discussion about the function \chi(C) has been clarified and extended in the second version of the manuscript submitted to SciPost, see for instance (4.20) or (4.26) and the discussion around them. Expression (4.26) explicitly gives an equation to determine the function \chi(C).

This function is determined by imposing initial conditions for the wave function as in an ordinary problem in quantum mechanics, where the wave function at t=0 has to be prescribed in order to have the full time dependent state.

NOTE THAT AS WE ARE COMPUTING WAVE FUNCTIONS THE FUNCTION \chi(C) CAN BE NEGATIVE OR EVEN COMPLEX. THIS IS BECAUSE IN THE AMPLITUDE THIS FUNCTION IS NOT INTERPRETED AS A DENSITY OF STATES.

Below equation (4.26) you can find the sentence

Solving the equation (4.26) for χ(C), complete determines the time dependent state.

The origin of this function is well explained in the footnote in the same page.

I would like to add one more comment about this part:

[ ...χ(C) that is undetermined by the Wheeler-de Witt equation.]

I It is true that the function \chi(C) is undetermined by the Wheeler-de Witt equation. However, in my work the Wheeler-de Witt equation does not hold as it stands.

Note please that the Wheeler-de Witt equation holds on manifolds with compact or open spatial sections. On these manifolds we are not allowed to fix the induced metric at the spatial boundaries (simply because there are not boundaries), so no particular proper time can be defined, so there is not way to introduce time evolution.

In my approach as there are spatial boundaries, on them we fix the induced metric hence the proper time along them. The appearance of time converts the problem in an initial value problem where the function at some initial time has to be specified in order to fully find the state. This is an ordinary procedure in partial differential equations and quantum mechanics or quantum field theory. Different initial functions will lead to different time dependent states.

[Can χ(C) be determined by matching the amplitude to a quantity computed in JT gravity solely from the path integral? ]

As explained in the previous paragraph \chi(C), in the case of the amplitudes, is determined by imposing initial conditions, so the answer to this question is NO.

[Perhaps this can be done in the limit τ1,2, l1,2, ϕ→∞ and taking an analytic continuation to Euclidean signature.]

No, it can not be done. Definitively we can make an analytic continuation to Euclidean signature but the manifold we get, as far as the metric concerns, is not a disk. It is topologically equivalent to a disk but still is a polygon with 4 sides, with four non-differentiable points. We can also take this limit τ1,2, l1,2, ϕ→∞ but in our case we would not get the AdS_2, partition function. This is because it is computed on a disk with a smooth boundary. Clearly the polygon does not have an smooth boundary. Note that on the four corners the reparametrization symmetry of the boundary would be broken to those corner preserving reparametrizations. Recall that the Schwarzian theory one gets over the boundary of the disk for the case of AdS_2 boundary conditions is a consequence of the invariance under reparametrizations of the 2d gravity action on the boundary.

[Should the resulting amplitude match the partition function in JT gravity since the boundary conditions along the four segments become the same as those for an asymptotic AdS2 boundary? Can that be used to determineχ(C)?]

Short answer no. We can impose AdS_2 boundary conditions in the Lorentzian strip, as done in (4.34)-(4.39) to show that this case is contained in my approach which is a more general framework, see for instance the report 1. In fact AdS_2 boundary conditions is the only case (as explained around equ. (4.38) and in the last paragraph in this section) that could evolve unitarily. Note please that still, even for AdS_2 boundary conditions, the function \chi(C) is determine by imposing initial conditions for the wave function, and different choices of the function \chi(C) will lead to different time dependent states.

[More broadly, it is unclear whether the transition amplitude in gravity along a finite strip should be a unitary process. In particular, if one considers the partition function of 2d gravity on a manifold whose boundary has a finite proper length, the density of states obtained by the Laplace transform has (at least naively) support on states that have a non-zero imaginary part. As discussed in reference [6] in the paper under review, this is related to the choice of the function that is undetermined by the WdW equation χ(C)]

For amplitudes, as previously explained, the function \chi(C) is not interpreted as a density of states. So, this function in my approach can be negative o even complex, as in ordinary quantum mechanics. The choice of the set of number this function belongs, from the point of view of amplitudes, does not play any role in the (non) unitary outcome. This has been stated in footnote 3 equ. (4.33) and complemented in footnote 4 equ. (4.40).

[Additionally, I believe the ideas presented in section 5 should be clarified. Is the author claiming that the results found, for instance, in (5.3), should be viewed as corrections to the presumably leading transition amplitude found in section 4?
The two geometries satisfy different boundary conditions (the one in section 4 has timelike boundaries while the geometry studied in section 5 does not), so that should not be the case.
It is indeed the case that there are geometries with other topologies that satisfy the same boundary conditions that contribute to the transition amplitude. These could be analytic continuations of the handle disk or of other higher-genus geometries; nevertheless, the author does not compute their contribution.
Moreover, it is unclear whether the calculation (5.3) captures the contribution of a hyperbolic cylinder or a hyperbolic annulus. Namely, does the geometry described by (5.3) have a closed geodesic as on the cylinder, or does it not as on the annulus? Is that again determined by the choice of χ(C)?]

Section 5 starts like this:

In what follows we will see how by gluing the two spatial boundaries...

This is indicating the reader in what sense we will obtain new topologies. As we are gluing the two spatial boundaries, namely, integrating over the boundary values (or integrating over the proper times), it is clear that after the gluing process any reference to the Lorentzian proper time on the boundaries disappears. In fact after gluing there are not spatial boundaries anymore. Only the boundaries in time remain, hence the resulting geometry is topologicaly a cylinder.

Then I explain the gluing process to get equ. (5.1). After equ. (5.2) we speculate about the existence of the function \chi(C) for the cylinder propagator, but we clearly stated that we have not been able to fully perform such calculation. Then we move to the geometry that is obtained by shrinking one of the cylinder ends, leading to equ. (5.3). So, equ. (5.3) can not capture the contribution of the hyperbolic cylinder simply because it is not its wave function. Equ. (5.3) corresponds to the wave function of a cap manifold. This is why in equ. (5.3) it is only needed the boundary values at just one boundary.

I hope all these answers and comments help the referee to have a better understanding of my work. Thank you very much again for this great opportunity.

---

## Round 3 · Referee Report · Anonymous (Referee 1) · 2023-11-15

Strengths

1. Considers JT gravity in the presence of spacelike plus timeline boundaries plus corners, which is more general than most existing analyses

2. Discusses quantum aspects, including arbitrary topologies

Weaknesses

1. A sizable fraction of the paper reviews material from the 1990ies; while this is okay for the sake of self-containment, it highlights that there is only a modest amount of new material in the paper.

2. There are open questions concerning the sections with new material, sections 4-6 (see my report).

Report

In my previous report, I have been focused on the review part of the paper and one of my main points was to clarify that the first 3 sections do not contain new results. This is addressed adequately in the amended manuscript.

In my current report, I focus on the last 3 sections, which contain novel results. It is clear that the paper stands or falls depending on the veracity of these 3 sections. I have some concerns with these sections, and the author should clarify them before the paper can be published:

Note: My convention is to denote minor points by -) and major points by *)

Section 4:

*) At the beginning of section 4, there is a slightly mysterious remark: the author states that performing the path integral over the dilaton seems to lead to troubles. I am not sure what the author means precisely: path integration over the dilaton would enforce a functional delta function, imposing the constraint that curvature is constant (and, for the usual choice nowadays, negative). What exactly is problematic about this?

*) Relatedly, the author remarks that he is unsure whether or not classical solutions are compatible with the boundary conditions for generic choices of some variables. Perhaps this remark is an attempt to explain the previous point. I did not get the gist of what the author was trying to say here. On general grounds, when imposing boundary conditions, we need to make sure that there are at least some classical field configurations compatible with these boundary conditions since otherwise there will be no saddle point contribution to the path integral, and the bulk and boundary equations of motion clash with each other. If this were the issue then the plausible conclusion would be to drop the boundary conditions as unsuitable. However, I do not think that this is what the author wants to convey. So some clarification by the author would be appreciated.

-) After (4.2) the author mentions the path integral measure and that he is not going to solve the path integral explicitly. Again, I am tempted to point back to the 1990ies, where the path integral for all 2d dilaton gravity models has been performed (though not necessarily with the boundary conditions imposed in the present work). So my question is, did the author not perform the path integral because it is somehow irrelevant for the purpose of the paper? (but why? Would not performing the path integral illuminate quantum aspects?) Or did the author shy away from it because it seemed too complicated?

*) The boundary condition (4.7) is overly restrictive. It freezes the dilaton along the boundary, and while this is good enough for some aspects of 2d dilaton gravity (like black hole thermodynamics or obtaining constant representatives of some coadjoint orbits), it is too restrictive for holographic applications and any application interested in boundary excitations (e.g., in an SYK context). Indeed, one can show in full generality that all the boundary charges vanish for any choice of boundary conditions that involve fixing the dilaton at the boundary. While this choice is not inconsistent, it makes the classical and quantum theory relatively boring. So let me ask bluntly: why did the author assume (4.7), and can the results be generalized when dropping (4.7)? Conversely, having imposed (4.7) the author eliminates all boundary excitations. So why is this an interesting choice for physics applications?

-) After (4.12), the author mentions "factor ordering." Perhaps this is formulation dependent or it only applies to the bulk theory and not to the boundary terms, but let me point out that there is no operator ordering ambiguity in the bulk for generic 2d dilaton gravity. So does this statement arise because of boundary/corner contributions? If yes, it could be helpful to point this out.

-) At the beginning of section 4.1 the author shows his intent to solve the path integral. Given this intention, how are we supposed to understand the earlier dismissal of the path integral measure (see three items above)?

*) After (4.19) it is stated that (4.18) is "complex for most of the values of C and \lambda". I find this remark a bit misleading. First of all, the sign of \lambda is fixed from the very start (we may assume AdS, so negative \lambda). Assuming this, the last term under the square root in (4.18) is always positive and hence there is no issue. Second, the sign of C is the sign of the mass (btw, the author never mentions this interpretation of C as mass - at least I did not find the words "mass" or "energy" anywhere in the paper). So depending on that sign different scenarios are possible: either the square root remains real for all values of the dilaton, or there is a lowest value of the dilaton (the locus of the black hole horizon). The author should considerably expand on the statement after (4.19), maybe along the lines above, rather than tersely remarking that (4.18) is complex.

*) After (4.21) it is stated that the function \chi(C) can be fixed by demanding initial conditions, but not how it is fixed precisely - what is the functional form of \chi(C) once this fixing is implemented? Phrased differently, how precisely is (4.26) solved for \chi(C)? In that equation, it appears as a function in an integral. But there are infinitely many functions that integrate to the same number, so it is unclear to me how (4.26) determines \chi(C).

-) After (4.32) the author stresses that he is not developing a mini-superspace model of JT gravity. I do not understand how this statement can make sense, at least for a standard interpretation of mini-superspace. My point is the bulk JT phase space has zero dimensions - there is no local propagating physical degree of freedom, so what does "mini-superspace" even mean for a topological field theory where everything is locally pure gauge?

-) Further below (4.32), I find the presentation about (non-)unitarity a bit confusing. The start sounds like a sweeping statement that makes the reader expect generically non-unitarity. But then, after (4.34)-(4.36), it becomes clear that these sweeping statements apply to dS, not to AdS. I think the author should be more upfront about this distinction between dS and AdS.

-) Regarding the counterterm (4.39), in general, it is neither necessary nor sufficient to have a finite on-shell action, so "making the action well-defined", while incidentally yielding the correct result, is in general not the correct procedure to get the counterterm. Instead, the correct procedure is holographic renormalization, which makes the variational principle well-defined. Since in this example both approaches yield the same result I'll let this slide, but still wanted to point it out to the author for future purposes. In any case, the result (4.39) is well-known and should be referenced.

Section 5:

*) Some of the key issues are related to my question above concerning \chi(C) and how it is determined. Eq. (5.4) seems to come out of the blue (or rather, from earlier work by others) - is there any way this result can be derived given the results in the current paper? If not, this seems a major drawback of the current approach.

-) After (5.6) the author should be more explicit. Are (5.5) and (5.6) in agreement with [7]? If yes, then please state this clearly. If no, what is the difference and why is there any difference to begin with? Also, it would be nice to define the functions H_2^(1) and K_2 somewhere. (I assume they are Hankel and Bessel functions of the second kind, so why not just mention this here?)

After clarifying these points the paper may be suitable for publication in SciPost Physics.

Requested changes

The requested changes are implied by my item list in the report.

  • validity: ok
  • significance: high
  • originality: high
  • clarity: good
  • formatting: perfect
  • grammar: good

Author:  Jose Alejandro Rosabal Rodriguez  on 2024-03-06  [id 4345]

(in reply to Report 2 on 2023-11-15)
Category:
remark
answer to question

Dear referee

Thanks for the second part of the report. This time I should say that there are fair questions but others show some degree of conceptual issues. I have to mention that I would have been very grateful if you had sent a unique and full report focusing on the whole paper, instead of two reports with several moths between them, and each of them focusing in half of the manuscript. In what follows I am going to provide the answers and address some of the conceptual issues in the questions.

MY ANSWERS , WILL APPEAR AFTER THE REFEREE QUESTIONS DENOTE BY -) OR *) AND SOMETIMES TO EMPHASIZE SOMETHING I WILL USE CAPITAL LETTERS.

Section 4:

*) At the beginning of section 4, there is a slightly mysterious remark: the author states that performing the path integral over the dilaton seems to lead to troubles. I am not sure what the author means precisely: path integration over the dilaton would enforce a functional delta function, imposing the constraint that curvature is constant (and, for the usual choice nowadays, negative). What exactly is problematic about this?

*) Relatedly, the author remarks that he is unsure whether or not classical solutions are compatible with the boundary conditions for generic choices of some variables. Perhaps this remark is an attempt to explain the previous point. I did not get the gist of what the author was trying to say here.

In section 4, equ. (4.2), the action is written in the Hamiltonian form. Upon exploring (4.1) it is clear that integrating the dilaton first is not an option. Even if one insists with the dilaton integration first, in the Hamiltonian formalism this path integration does not enforce any functional delta function.

Nonetheless, when the action is written in the Lagrangian form, and the boundary conditions are those corresponding to a constant curvature space (dS or AdS), It is suitable to integrate the dilaton first because its equation of motion is compatible with those boundary conditions. When different boundary conditions to those of constant curvature space are used, dilaton integration is not an option any more because its equation could be not compatible with the prescribed boundary conditions.

This does not mean that we can not solve JT gravity with different boundary conditions. It means that the techniques used in the modern context of JT (modern in the sense that it is linked with SYK and Holography, with AdS_2 boundary conditions) are not applicable to this more general case developed in the manuscript. In fact this is why there are just a few references concerning this modern context of JT gravity in the paper, simply because we did not need them to develop the approach presented in the manuscript.

Let us stress that when posing a mathematical problem we first fix the boundary conditions and they can be generic. Then we ASK to the theory whether or not these boundary conditions are compatible with its dynamics. If the theory is classical, there will not be classical solutions for those boundary conditions that are not compatible with the dynamics of the theory. In a quantum theory this answer comes from the amplitudes. If a process can occur for some boundary conditions then the corresponding amplitude (or probability) will be different from zero, and zero otherwise. We never pose a mathematical problem in the other way around. Namely, we do not chose the theory first and then let it to decide the boundary conditions for us.

It could be the case that for some boundary conditions there are not classical solutions but there could be quantum transitions.

*) On general grounds, when imposing boundary conditions, we need to make sure that there are at least some classical field configurations compatible with these boundary conditions

We partially agree with this, one WOULD like to be sure that the boundary conditions one wants to impose are compatible with the dynamics of the theory. However, how can one know this in advance? To be sure, you have to impose the boundary conditions and ASK to the theory whether or not they are GOOD boundary conditions. You can only get an answer by solving the equation of motion or the amplitude, depending on the context.

*) since otherwise there will be no saddle point contribution to the path integral, and the bulk and boundary equations of motion clash with each other.

If there were not classical solutions and we were working SEMICLASSICALLY the path integral would be zero, that is correct. Note however that we are not working semiclassically. Just in case, we need to mention that not only the saddle points contribute to a path integral. In general all possible field configurations contribute to a path integral.

*) If this were the issue then the plausible conclusion would be to drop the boundary conditions as unsuitable. However, I do not think that this is what the author wants to convey. So some clarification by the author would be appreciated.

RECALL THAT WE ARE WORKING NON PERTURBATIVELY. SO, THE PATH INTEGRATION IS OVER ALL POSSIBLE FIELD CONFIGURATIONS. IF FOR SOME BOUNDARY CONDITIONS THERE ARE NOT CLASSICAL SOLUTIONS IT ONLY MEANS THAT THERE ARE NOT CONTRIBUTION FROM THE CLASSICAL SECTOR TO THE PATH INTEGRAL. HOWEVER THE NON CLASSICAL SECTOR IN THE SPACE OF FIELDS DOES CONTRIBUTE TO THE PATH INTEGRAL.

-) After (4.2) the author mentions the path integral measure and that he is not going to solve the path integral explicitly. Again, I am tempted to point back to the 1990ies, where the path integral for all 2d dilaton gravity models has been performed (though not necessarily with the boundary conditions imposed in the present work). So my question is, did the author not perform the path integral because it is somehow irrelevant for the purpose of the paper? (but why? Would not performing the path integral illuminate quantum aspects?) Or did the author shy away from it because it seemed too complicated?

Perhaps we were not clear enough in this statement. Clearly the path integral has been solved to get the amplitude. What we wanted to say is that we do not use the conventional methods to solve the path integral and we do not need to perform an actual path integration. In the procedure presented in the manuscript using a canonical transformation we were able to reduce the action in such a way that no functional integration has to be performed to get the final amplitude. This will be clarified in the new version.

*) The boundary condition (4.7) is overly restrictive. It freezes the dilaton along the boundary, and while this is good enough for some aspects of 2d dilaton gravity (like black hole thermodynamics or obtaining constant representatives of some coadjoint orbits), it is too restrictive for holographic applications and any application interested in boundary excitations (e.g., in an SYK context). Indeed, one can show in full generality that all the boundary charges vanish for any choice of boundary conditions that involve fixing the dilaton at the boundary.

(4.7) is not too restrictive as the referee claims. Certainly, one can have a varying dilaton over the spatial boundaries but for this case less conclusion can be made related to the amplitude.

A good reference the referee can take a look to see the applications of this boundary conditions is 2301.07257 equ. (2.6), see also references there in, (I am including these references in the new version). We did not know this paper before. What is interesting about it is that the authors develop JT gravity on an INFINITE (in time) Lorentzian strip (two boundaries in space) with AdS_2 and constant dilaton boundary conditions. Of course, with AdS_2 bounady conditions much more conclusions can be made regarding the quatization of JT gravity. For the purpose of comparison we just want to mention that we develop JT gravity on a FINITE (in time) Lorentzian strip (two boundaries in space), with GENERIC boundary conditions.

*) While this choice is not inconsistent, it makes the classical and quantum theory relatively boring.

Perhaps the referee finds BORING my calculations because she/he is out of context. Please take a look at 2301.07257 and references there in for several other applications of this boundary condition?

*) So let me ask bluntly: why did the author assume (4.7), and can the results be generalized when dropping (4.7)?

LET US EMPHASIZE AGAIN THAT (4.7) IS JUST A CHOICE TO COMPARE OR CONTRAST WITH SOME KNOWN RESULTS, THE DILATON IN MY APPROACH CAN TAKE ANY VALUE YOU WISH ON THE BOUNDARIES. THIS IS WHAT WE MEAN BY GENERIC BOUNDARY CONDITIONS. FOR INSTANCE IN SEC. 5 IN ORDER TO MAKE NEW TOPOLOGIES WE NEED TO ASSUME A VARYING DILATON OVER THE SPATIAL BOUNDARIES.

IN FACT, IF THE DILATON WERE CONSTANT WE COULD NOT GLUE THE TWO SPATIAL BOUNDARIES. THE GLUING PROCESS INVOLVE A TRACE OVER ALL THE BOUNDARIES VALUES. IF THE DILATON WERE CONSIDERED CONSTANT OVER THE BOUNDARY CERTAINLY WE WILL NOT BE TRACING OVER ALL BOUNDARY VALUES.

I THINK THIS IS THE BEST MOTIVATION TO POSE A PROBLEM WITH GENERIC BOUNDARY CONDITIONS. THOSE STUDIES WHERE THE BOUNDARY CONDITIONS ARE LIMITED TO CONSTANT CURVATURE SPACE ARE LIMITED TO CREATE NEW TOPOLOGIES BY GLUING ITS BOUNDARIES. MORE IMPORTANTLY THEY ARE LIMITED IN THE KIND OF STATE OR LORENTZIAN AMPLITUDES THEY CAN COMPUTE, IF ANY CAN BE COMPUTED.

*) Conversely, having imposed (4.7) the author eliminates all boundary excitations. So why is this an interesting choice for physics applications?

The referee already answered this question in the beginning of this item, and in the reference above more physical applications can be found.

NONETHELESS, WE ARE NOT INTERESTED IN HOLOGRAPHY... AND OUR RESULTS DO NOT RELAY ON THE EXISTENCE OF THE ADS/CFT CORRESPONDENCE.

WE ARE ONLY INTERESTED IN COMPUTING TIME DEPENDENT AMPLITUDES IN PURE QUANTUM GRAVITY TO ATTACK THE INFORMATION PARADOX FROM A DIFFERENT PERSPECTIVE. AFTER ALL IT HAS BEEN 40 YEARS OF COMPUTING ENTROPIES BUT SO FAR THE INFORMATION PARADOX HAS NOT BEEN CRACKED. IN FACT THE ORIGINAL IDEA OF HAWKING WAS TO COMPUTE AMPLITUDES AND NOT ENTROPIES SEE [1]. THE ISSUES WAS THAT AT THAT TIME NOT ENOUGH TOOLS HAD BEEN DEVELOPED TO COMPUTE THEM; AND WITH OUT ANY DOUBT IT IS EASIER TO COMPUTE AN ENTROPY THAN AN AMPLITUDE. THIS IS WHY ENTROPY CALCULATIONS BECAME MORE POPULAR. IT IS TIME TO GIVE THE AMPLITUDES ONE MORE OPPORTUNITY.

-) After (4.12), the author mentions "factor ordering." Perhaps this is formulation dependent or it only applies to the bulk theory and not to the boundary terms, but let me point out that there is no operator ordering ambiguity in the bulk for generic 2d dilaton gravity. So does this statement arise because of boundary/corner contributions? If yes, it could be helpful to point this out.

I am not sure in what reference the referee found the statement that 2d dilaton gravity does not have this ambiguity. For sure we can point out reference [12] where it is well explained why ALL 2d dilaton gravity needs a factor ordering prescription. You can see also [9] and in the new version some of the new references too.

In fact, a quick exploration of (4.12) shows that it is not a solution of the naive Wheeler-DeWitt equation one gets, for instance, from (3.15).

-) At the beginning of section 4.1 the author shows his intent to solve the path integral. Given this intention, how are we supposed to understand the earlier dismissal of the path integral measure (see three items above)?

Like I said above. I do solve the path integral. The methods I use allow me to reduce the path integral in such a way that in the end no actual path integration is needed.

*) After (4.19) it is stated that (4.18) is "complex for most of the values of C and \lambda". I find this remark a bit misleading. First of all, the sign of \lambda is fixed from the very start (we may assume AdS, so negative \lambda). Assuming this, the last term under the square root in (4.18) is always positive and hence there is no issue. Second, the sign of C is the sign of the mass (btw, the author never mentions this interpretation of C as mass - at least I did not find the words "mass" or "energy" anywhere in the paper). So depending on that sign different scenarios are possible: either the square root remains real for all values of the dilaton, or there is a lowest value of the dilaton (the locus of the black hole horizon). The author should considerably expand on the statement after (4.19), maybe along the lines above, rather than tersely remarking that (4.18) is complex.

What we wanted to say is that (4.18) is complex (or imaginary) for most of the values of C ( C ranges from zero to infinity) and \phi, for a fixed \lambda.

For those familiar with JT gravity it is well known that C is a mas parameter, hence C>=0. This manuscript is aimed to a specialized reader this is why I did not make any comment about the physical interpretation of C. This will be clarified.

Using the following two sentences I will show how wrong the referee is this regard.

"First of all, the sign of \lambda is fixed from the very start (we may assume AdS, so negative \lambda). Assuming this, the last term under the square root in (4.18) is always positive and hence there is no issue."

IT SEEMS THERE IS NO PROBLEM IN THIS CASE, HOWEVER THE REFEREE NEEDS TO THINK ABOUT THIS ACTION INSIDE THE PATH INTEGRAL. AS C IS THE NEW DEGREE OF FREEDOM WE HAVE ONE INTEGRATION IN C. AS C RANGES FROM ZERO TO INFINITY THERE WILL BE A VALUE OF C (NO MATTER HOW SMALL LAMBDA IS, PROVIDED IT IS FINITE) FROM WHICH (4.18) BECOMES IMAGINARY. CERTAINLY AS C RANGES FROM ZERO TO INFINITE THESE ARE MOST OF THE VALUES OF C, RIGHT? THE REFEREE CAN REPEAT THIS EXERCISE AND GETS CONVINCE THAT (4.18) IS IMAGINARY FOR MOST OF THE VALUES OF C AND \PHI. EXCEPT WHEN \PHI AND \VAR[N] GO TO INFINITE AS IN (4.34)-(4.36). IN THIS CASE (4.18) IS REAL NO MATTER THE VALUE OF C. THE PRICE TO PAY IN THIS CASE IS THAT THE ACTION NEEDS REGULARIZATION.

*) After (4.21) it is stated that the function \chi(C) can be fixed by demanding initial conditions, but not how it is fixed precisely - what is the functional form of \chi(C) once this fixing is implemented? Phrased differently, how precisely is (4.26) solved for \chi(C)? In that equation, it appears as a function in an integral. But there are infinitely many functions that integrate to the same number, so it is unclear to me how (4.26) determines \chi(C).

(4.26) determines the function \chi(C) as in ordinary quantum mechanics a similar expression determines the Fourier coefficients of a given general solution of the Schrodinger equation. This Schrodinger equation solution can be found by solving the differential equation with whatever method, or computing the corresponding path integral. In ordinary quantum mechanics to fully specify a state at some time t>0, ONE NEEDS TO PRESCRIBE THE STATE AT SOME t=0, see for instance the Feynman papers on path integral, or any serious book on quantum mechanics or QFT.

(4.26) can be seen as a generalization of the statements in previous paragraph. note that it is a FUNCTIONAL relation and the function \chi(C) plays the role of the Fourier coefficients. Note that the right hand side is a given functional, namely the PRESCRIBED initial state at t=0. In other words, as in ordinary quantum mechanics or QFT, HERE one needs to PRESCRIBE THE STATE at t=0 in order to fully determine the time dependent amplitude we are interested in.

In addition the discussion between (4.32) and (4.34) together with the footnote 3 complement the discussion around (4.26). Here the referee can see that in order to fully and consistency determine the function \chi(C) we need to know the product in the Hilbert space, as in ordinary quantum mechanics. The product between states in this more general setting will be found elsewhere. We remark that not knowing the product between state explicitly does not prevent us to make conclusions about the (non) unitary behavior of the amplitude, see for instance appendix C of 2309.03639.

In the new version I will expand this discussion.

-) After (4.32) the author stresses that he is not developing a mini-superspace model of JT gravity. I do not understand how this statement can make sense, at least for a standard interpretation of mini-superspace. My point is the bulk JT phase space has zero dimensions - there is no local propagating physical degree of freedom, so what does "mini-superspace" even mean for a topological field theory where everything is locally pure gauge?

WE ASSUME THAT THE REFEREE CONFUSION IS ABOUT THE TERM MINI-SUPERSPACE IN THE CONTEXT OF JT GRAVITY AND NOT RELATED TO WHAT WE DID NOT DO IN THE PAPER. WE ARE SAYING THAT WE ARE NOT (NOT) DEVELOPING SUCH A MODEL.

THE COMMENT IS IN ORDER AROUND THIS EQUATION BECAUSE EQU (4.32) LOOKS SIMILAR TO THE EQUATION ONE GETS IN A MINI-SUPERSPACE MODEL OF JT GRAVITY.

Again this paper is developed for readers that are familiar with JT gravity but with quantum gravity in general too. In some models of JT gravity one can find a similar equation to (4.32) but for L in the Reals, see for instance reference [9] and references there in, see also section 2.4 of [7]. Usually these models are called MINI-SUPERSPACE MODELS (YOU CAN FIND THE WORD IN THESE REFERENCES AND MANY MORE) because their metric functions are only time dependent. This is why one gets a Wheeler-DeWitt equation that is an ordinary differential equation instead of a functional differential equation. In contrast for the so called midi-superspace models the metric functions depend on time and one spatial coordinate and the Wheeler-DeWitt equation is a functional differential equation.

Regarding the number of propagating degrees of freedom, it is true that there are not propagating degrees of freedom in the bulk for JT gravity. This is not the case when the manifold has boundaries, as in the manuscript for the finite strip. When there are spatial boundaries, in the Lorentzian case, some of the degrees of freedom can propagate along them.

-) Further below (4.32), I find the presentation about (non-)unitarity a bit confusing. The start sounds like a sweeping statement that makes the reader expect generically non-unitarity. But then, after (4.34)-(4.36), it becomes clear that these sweeping statements apply to dS, not to AdS. I think the author should be more upfront about this distinction between dS and AdS.

I AGREE, I WILL MODIFY AND EXPAND THESE DISCUSSIONS.

-) Regarding the counterterm (4.39), in general, it is neither necessary nor sufficient to have a finite on-shell action, so "making the action well-defined", while incidentally yielding the correct result, is in general not the correct procedure to get the counterterm. Instead, the correct procedure is holographic renormalization, which makes the variational principle well-defined. Since in this example both approaches yield the same result I'll let this slide, but still wanted to point it out to the author for future purposes. In any case, the result (4.39) is well-known and should be referenced.

WE PREFER DO NOT SLIDE THIS POINT. WE WANT TO REPEAT THAT WE ARE NOT INTERESTED IN HOLOGRAPHY, SO I DO NO HAVE TO APPLY THE TECHNIQUES OR EXPLANATIONS COMING FROM IT TO MY WORK.

FORTUNATELY EVERYTHING HERE CAN BE EXPLAINED WITHIN PURE GRAVITY. ALTHOUGH THE REFEREE CAN CALL IT HOLOGRAPHIC RENORMALIZATION TO THE PROCEDURE (4.35)-(4.40) IF SHE/HE WISHES, NOBODY DOES IT. AFTER ALL THE PROCEDURE (4.35)-(4.40) IS SIMILAR TO WHAT ONE DOES IN HOLOGRAPHIC RENORMALIZATION. HOWEVER IT IS WORTH TO MENTION THE PROCEDURE IN THE MANUSCRIPT HAS BEEN IN THE QG LITERATURE WAY BEFORE HOLOGRAPHY AND HOLOGRAPHIC RENORMALIZATION. THE REFEREE CAN TAKE A LOOK AT 1606.01857, 1904.01911, 2301.07257. (6,7,8 IN THE NEW VERSION) AND GETS CONVINCED THAT THEY DO NOT USE SUCH A TERM IN THE CONTEXT OF PURE GRAVITY, EVEN WITH ADS_2 BOUNDARY CONDITIONS.

IN THE CONTEXT OF PURE GRAVITY THE COUNTER TERM HAS BEEN COMPUTED FOLLOWING THE USUAL ARGUMENTS WHEN A QUANTITY DIVERGES. THE REFEREE CAN FIND SIMILAR PROCEDURE IN SEVERAL PAPERS ON QUANTUM GRAVITY.

IN FACT I INVITE THE REFEREE TO CHECK THE VARIATIONAL PRINCIPLE WITH (4.39) AND THE ACTION (3.14) WITH GENERIC BOUNDARY CONDITIONS ON A FINITE LORENTZIAN STRIP. LET US POINT OUT ALSO EQU. (3.3) PERHAPS THE REFEREE WILL NEED IT IN ORDER TO CHECK THAT THE VARIATIONAL PRINCIPLE DOES WORK.

IT HAS BEEN PROPERLY REFERENCED, SEE [6] BETWEEN (4.34) AND (4.35) IN THE OLD VERSION. IN THE NEW VERSION YOU CAN FIND THREE REFERENCES.

Section 5:

*) Some of the key issues are related to my question above concerning \chi(C) and how it is determined. Eq. (5.4) seems to come out of the blue (or rather, from earlier work by others) - is there any way this result can be derived given the results in the current paper? If not, this seems a major drawback of the current approach.

(5.4) is a particular choice that one has allowed to do. When an initial state is prescribed we do not need to derive it from anywhere. The theory tells us that we are allowed to prescribe any, (ANY) initial state, this is basic in quantum mechanics also in math. So, (5.4) has been chosen just to contrast with known results. Equ. (4.26) tells us that basically we can choose \chi[C] as we wish. Te result of the integration, namely the final form of the amplitude will tell us if the choice of \chi[C] (recall that this is equivalent to prescribe the initial state) was a GOOD choice or not.

-) After (5.6) the author should be more explicit. Are (5.5) and (5.6) in agreement with [7]? If yes, then please state this clearly. If no, what is the difference and why is there any difference to begin with? Also, it would be nice to define the functions H_2^(1) and K_2 somewhere. (I assume they are Hankel and Bessel functions of the second kind, so why not just mention this here?)

I will expand and clarify this discussion. (5.5) is similar but it does not coincide with [7] because we are using the contracting branch but they are using the expanding one.

After clarifying these points the paper may be suitable for publication in SciPost Physics.

WE BELIEVE THAT THE REFEREE IS CONFUSING SEVERAL ASPECTS OF JT GRAVITY. WE WOULD LIKE TO MENTION THAT DEPENDING ON THE MANIFOLD (METRIC SIGNATURE AND TOPOLOGY) JT GRAVITY MIGHT PRESENT DIFFERENT FEATURES. THE REFEREE CAN SEE THE EXAMPLES IN THE LITERATURE.

THE PAPER HAS BEEN PREPARED AND WRITTEN IN A PURE QG LANGUAGE AND TAKING DISTANCE FROM HOLOGRAPHY BECAUSE WE WANTED TO HAVE A SUITABLE PLATFORM FOR A GENERALIZATION TO 4D GRAVITY BEYOND HOLOGRAPHY, SEE 2309.03639. THE MODERN APPROACH OF JT GRAVITY, AND ITS CORRESPONDENCE TO SYK, AND THE SCHWARZIAN THEORY IS A LIMITED FORMULATION IN THE SENSE THAT IS FAR FROM BEING SUITABLE FOR A GENERALIZATION TO 4D. DOES THE REFEREE AGREE?

SO FAR WE HAVE LEARNT NOTHING FROM JT GRAVITY THAT CAN BE APPLIED TO 4D GRAVITY. MY PAPER MIGHT PROVIDE SOME HITS ON HOW TO TREAT QG IN 4D, EVEN INCORPORATING TIME.

THANK THE REFEREE AGAIN BECAUSE AFTER THESE QUESTIONS AND THIS REVISION THE MANUSCRIPT LOOKS MUCH MORE ROBUST.

---

## Round 3 · List of Changes

1- New references added, one of them suggested by the first referee, it has been properly cited.

2- Several typos corrected and new comments added.

3-Some discussions has been extended as well as the acknowledgments.

4- Several changes regarding the function $\chi(\text{C})$, and a considerably expanded discussion about it has been including.

---

## Editorial Decision

in_refereeing